# Enhanced single RNA imaging reveals dynamic gene expression in live animals

Yucen Hu[1†], Jingxiu Xu[2†], Erqing Gao[2†], Xueyuan Fan[1], Jieli Wei[1], Bingcheng Ye[1], Suhong Xu[2,3]*, Weirui Ma[1]*

[1]Zhejiang Provincial Key Laboratory of Cancer Molecular Cell Biology, Life Sciences Institute, Zhejiang University, Hangzhou, China; [2]International Biomedicine-X research center of the Second Affiliated Hospital, Zhejiang University, Hangzhou, China; [3]Center for Stem Cell and Regenerative Medicine and Department of Burn and wound repair of the Second Affiliated Hospital, Zhejiang University School of Medicine, Hangzhou, China

**Abstract** Imaging endogenous mRNAs in live animals is technically challenging. Here, we describe an MS2-based signal amplification with the Suntag system that enables live-cell RNA imaging of high temporal resolution and with 8xMS2 stem-loops, which overcomes the obstacle of inserting a 1300 nt 24xMS2 into the genome for the imaging of endogenous mRNAs. Using this tool, we were able to image the activation of gene expression and the dynamics of endogenous mRNAs in the epidermis of live *C. elegans*.

## Editor's evaluation

The authors have amplified the signal on MS2 so that a smaller insertion is sufficient to track mRNA in vivo. They provide solid evidence that this approach generates a sufficient signal, equivalent to the full-length MS2. This work, along with a previously reported similar method will be useful to investigators considering single-molecule imaging in Elegans as well as other organisms.

*For correspondence:
shxu@zju.edu.cn (SX);
maweirui@zju.edu.cn (WM)

[†]These authors contributed equally to this work

**Competing interest:** The authors declare that no competing interests exist.

## Introduction

RNAs relay genetic information from DNA to proteins or function by themself. Live-cell imaging of RNAs at a single-molecule level is crucial to uncovering their roles in gene expression regulation (*Buxbaum et al., 2015*). Various tools have been developed to visualize RNAs in live cells (*Braselmann et al., 2020*; *Le et al., 2022*), including RNA-binding protein–fluorescent protein approaches (*Bertrand et al., 1998*), CRISPR-based systems (*Nelles et al., 2016*; *Yang et al., 2019*), and those utilizing fluorophore–RNA aptamer pairs (*Chen et al., 2019*; *Paige et al., 2011*; *Sunbul et al., 2021*). The MS2-based system is the most widely used and represents the current gold standard for single-molecule RNA imaging in live cells (*Braselmann et al., 2020*; *Bertrand et al., 1998*). MS2 is a short RNA stem-loop bound specifically by the bacteriophage MS2 coat protein (MCP). To image RNA, 24xMS2 are placed at the 3′UTR or 5′UTR, and a fluorescent protein is fused to the MCP (MCP-FP). When coexpressed in cells, up to 48 fluorescent proteins (2 × 24, with two MCPs bound to one MS2) will be recruited to the RNA through MS2–MCP binding. This forms a fluorescent spot indicating a single RNA molecule (*Braselmann et al., 2020*).

The MS2 system has been successfully used to trace the whole mRNA life-cycle from transcription, to nuclear export, subcellular localization, translation, and to final degradation (*Braselmann et al., 2020*; *Le et al., 2022*). However, most RNA imaging studies in animal cells have been performed using exogenous mRNAs in cultured cell lines. 24xMS2 (about 1300 nt in length) have to be knocked into a

specific genomic locus to image endogenous mRNA. The difficulty and low efficiency of the knock-in of long sequences into the genome represent a significant obstacle toward visualizing endogenous mRNA using the MS2 system. Thus, it is not surprising that less than 10 endogenous mRNAs have been imaged in live animal cells at a single-molecule level, and examples of endogenous mRNAs imaged in live animals remain extremely rare (*Halstead et al., 2015*; *Levo et al., 2022*; *Dufourt et al., 2021*; *Park et al., 2014*; *Das et al., 2018*; *Zimyanin et al., 2008*; *Forrest and Gavis, 2003*; *Lee et al., 2022*). Since overexpressed mRNAs may not faithfully recapitulate endogenous mRNA expression and dynamics, the development of more sensitive techniques for endogenous mRNA imaging is of great value.

## Results

In this study, we reasoned that combining the MS2 with a signal amplifier may allow the recruitment of more fluorescent proteins to the RNA with fewer MS2 repeats (i.e., 8xMS2 – see *Figure 1A*). To achieve this, we combined the MS2 and Suntag systems. Suntag is a 19 amino acid protein tag that binds to its specific single-chain variable fragment (scFv) antibody (*Tanenbaum et al., 2014*). We fused MCP with a 24xSuntag array and linked scFv with sfGFP. When coexpressed in cells, one MS2 interacts with two MCP-24xSuntag molecules, further recruiting 2 × 24 GFP molecules (*Figure 1A*). As the Suntag serves as a signal amplifier, the combined system was named as MS2-based signal Amplification with Suntag System (MASS). When an 8xMS2 is placed into the 3'UTR, up to 384 (2 × 8 × 24) GFP can then be tethered to a single mRNA through the MCP–Suntag–scFv–sfGFP interaction (*Figure 1A*). This leads to the formation of an intense GFP spot associated with single mRNA, facilitating live RNA imaging.

As proof of concept, 8xMS2 V1[4] was fused to the 3'UTR of *β-ACTIN* mRNA and transfected into HeLa cells. When all the required elements of the MASS (MS2, MCP, 24xSuntag, and scFv antibody) were present, bright GFP foci were readily detected (*Figure 1B*). As controls, no GFP foci were detected when omitting any one of these elements (*Figure 1B*). With MCP-24xSuntag, an MCP molecule could be labeled with up to 24 GFPs. Under our imaging conditions (100–500 ms exposure time), MCP-24xSuntag particles were not detected (*Figure 1B*), probably because MCP-24xSuntag are diffusing too fast to be imaged as a spot. Thus, the GFP foci clearly represent *β-ACTIN* RNA molecules.

In addition, we performed MASS combined with single-molecule RNA fluorescence in situ hybridization (smFISH) using a probe against the MS2 stem-loop and a probe against the linker region between the MS2 stem-loops (*Figure 1C*, *Figure 1—figure supplement 1A, B*). We found that MASS detected a similar number of GFP foci compared to the spots detected by smFISH (*Figure 1—figure supplement 1C*, *Figure 1—source data 1*). Moreover, the majority of GFP foci (72%) colocalized with the smFISH spots of *β-ACTIN-3'UTR-8xMS2* mRNAs (*Figure 1C, D*, *Figure 1—figure supplement 1B*, *Figure 1—source data 1*). It is reported that not all MS2 stem-loop will be bound by the MCP (*Wu et al., 2012*). As only 8xMS2 was used in MASS, it is likely that some mRNAs were not fully bound by MCP and were not detected. On the other hand, in theory, only up to 16 probes will be hybridized with the 8xMS2 stem-loops and the linker regions in the smFISH experiment, and it is possible that some mRNAs were miss labeled by smFISH. Therefore, 100% colocalization of MASS foci with the smFISH spots was hard to achieve. Taken together, our data indicated that MASS is able to detect single mRNA molecules and label the majority of mRNAs from a specific gene in live cells.

It was reported that tagging mRNAs with MS2 stem-loops might affect mRNA stability, which could be counteracted by using improved versions of MS2 repeats (*Li et al., 2022*; *Vera et al., 2019*). We found GFP foci of *β-ACTIN-3'UTR-8xMS2* mRNAs were detected regardless of using MS2-V1[4] or MS2-V7[21] with 2× tandem repeats of monomer MCPs (tdMCP) (*Figure 1—figure supplement 2A*). Therefore, MASS could be performed using different versions of MS2 stem-loops and MCPs. To directly test whether mRNA stability was affected while imaging by MASS, we examined the stability of three mRNAs: *MYC*, *HSPA1A*, and *KIF18B*, which were reported as medium stable mRNAs (*Vera et al., 2019*; *Sharova et al., 2009*). We found that the stability of those mRNAs was not significantly affected either by tagging of 8xMS2 V1 or by coexpression of the MASS imaging system (*Figure 1—figure supplement 2B, C*, *Figure 1—source data 1*). It is worth noting that we only examined three mRNAs in this study. The stability of specific mRNAs might be affected by MASS. If so, an improved version of MS2 should be used for the imaging experiment.

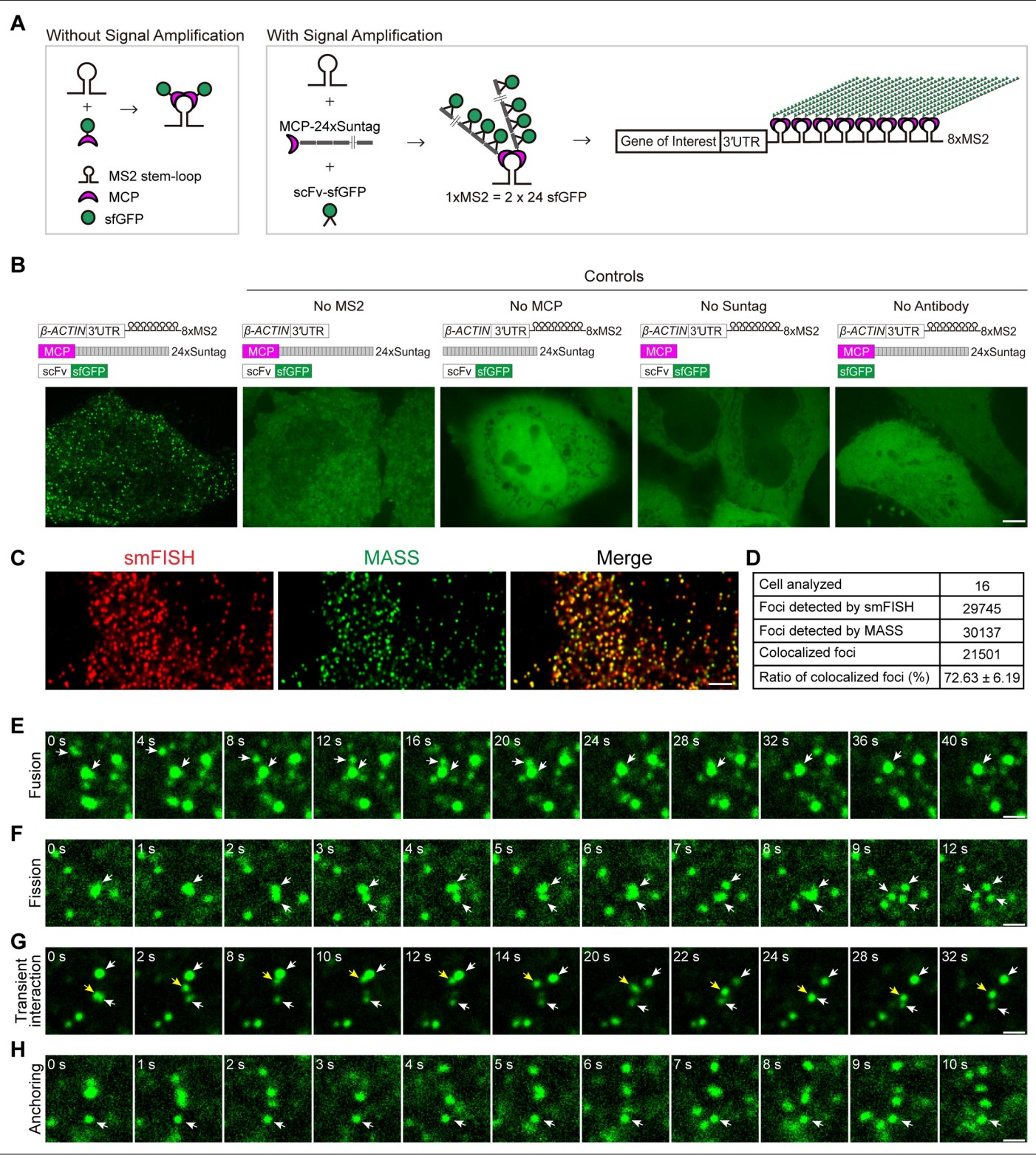

**Figure 1.** Live-cell imaging of *β-ACTIN* mRNA with the MS2-based signal amplification with the Suntag system. (**A**) Schematic of the classical MS2-MCP system and the MS2-based signal amplification with the Suntag system. (**B**) Representative images of *β-ACTIN-8xMS2* mRNA in live HeLa cells. The sfGFP fluorescence signal is shown. Left panel: Constructs of *β-ACTIN-8xMS2*, MCP-24xSuntag, and scFv-sfGFP were cotransfected into HeLa cells. Images were taken 12 hr after transfection. Right panels: Where one of the elements was removed (as indicated). Scale bar, 5 μm. (**C**) Representative confocal images of *β-ACTIN-8xMS2* labeled by MASS and single-molecule in situ hybridization (smFISH) with probes against MS2

*Figure 1 continued on next page*

*Figure 1 continued*

stem-loops and probes against the linker region between the MS2 stem-loops in HeLa cells. Scale bar, 5 µm. See also *Figure 1—figure supplement 1*. (**D**) Quantification of the total and colocalized foci of *β-ACTIN-8xMS2* mRNAs detected by smFISH or MASS with tdMCP-24xSuntag in HeLa cells. A total of 16 cells from three independent smFISH experiments were analyzed. See also *Figure 1—source data 1*. (**E–H**) Time-lapse imaging of *β-ACTIN-8xMS2* mRNA dynamics in HeLa cells. sfGFP foci (*β-ACTIN* mRNAs) are shown. Constructs of *β-ACTIN-8xMS2*, MCP-24xSuntag, and scFv-sfGFP were cotransfected into HeLa cells. Images were taken 12 hr after transfection. (**E**) A fusion event of two sfGFP spots (white arrows). (**F**) A fission event: with large sfGFP foci split into three spots (white arrows). (**G**) Transient interactions of an sfGFP spot (yellow arrow) between two spots (white arrows). (**H**) An sfGFP spot showing no movement over a 10-s period (white arrow). Scale bars, 1 µm.

The online version of this article includes the following source data and figure supplement(s) for figure 1:

**Source data 1.** qRT-PCR (Quantitative Reverse Transcription PCR), number of foci detected by smFISH and MASS, signal-to-noise ratio, velocity, and intensity of the foci of mRNA detected in HeLa cells.

**Figure supplement 1.** Labeling of *β-ACTIN-8xMS2* mRNA by smFISH and MASS with the tdMCP-24xSuntag in HeLa cells.

**Figure supplement 2.** MASS did not affect the stability of mRNAs.

**Figure supplement 3.** MASS did not affect mRNA subcellular localization.

**Figure supplement 4.** MASS has a higher signal-to-noise ratio than the conventional 24xMS2 method.

**Figure supplement 5.** MASS with tdMCP-6xSuntag did not affect the speed of mRNA movement.

**Figure supplement 6.** MASS labels mRNAs with a higher intensity than the conventional 24xMS2 imaging system.

It has been reported that *β-ACTIN* mRNAs with 3′UTR can localize to the lamellipodia (*Katz et al., 2012*). In support of this, we observed that the GFP foci of *β-ACTIN-3′UTR-8XMS2* mRNAs were indeed localized to the lamellipodia in HeLa cells (*Figure 1—figure supplement 3A* and *Video 1*). To further test whether mRNA localization was affected by MASS, we imaged *β-ACTIN-3′UTR* mRNA with MASS or with the conventional 24xMS2 system in NIH/3T3 cells, which is a mouse fibroblast cell line. We found that GFP foci of *β-ACTIN-3′UTR* mRNAs detected by MASS or 24xMS2 system showed similar localization (*Figure 1—figure supplement 3B*). Thus, these data suggested that MASS did not affect RNA subcellular localization.

Haven established that MASS with MCP-24xSuntag could image single RNA molecules; we next sought to test whether MASS could be performed with shorter repeats of Suntag arrays (MCP-12xSuntag, MCP-6xSuntag) and compare MASS to the conventional 24xMS2 image system.

With the conventional 24xMS2 mRNA imaging system, a nuclear localization signal (NLS) was usually fused to MCP to localize NLS-MCP-GFP into GFP. NLS-MCP-GFP will be exported into the cytoplasm with mRNAs when binding to mRNAs. This strategy allows the detection of *β-ACTIN-3′UTR* mRNAs with a signal-to-noise ratio of 1.21 (*Figure 1—figure supplement 4A, B*, *Figure 1—source data 1*). MASS with MCP-24xSuntag showed the highest signal-to-ratio of 1.79. MCP-12xSuntag and MCP-6xSuntag labeled *β-ACTIN-3′UTR*-8xMS2 mRNAs with a similar signal-to-noise ratio of 1.42 and 1.48, which are better than the conventional 24xMS2 system (*Figure 1—figure supplement 4C–I*, *Figure 1—source data 1*). Thus, MASS is flexible regarding the length of Suntag repeats used for imaging. MASS with short repeats of Suntag (MCP-6xSuntag) is sufficient for RNA labeling.

One critical concern about MASS is that intense tagging of mRNAs may affect the dynamics of mRNAs. To address this, we performed live-cell imaging of *β-ACTIN* mRNA using the conventional 24xMS2 system or MASS with different lengths of Suntag arrays (MCP-24xSuntag, MCP-12xSuntag, and MCP-6xSuntag). The velocity of mRNA movement in each imaging condition was measured. We found that compared to the conventional 24xMS2 system, mRNA labeled by MASS with MCP-24xSuntag or MCP-12xSuntag showed a

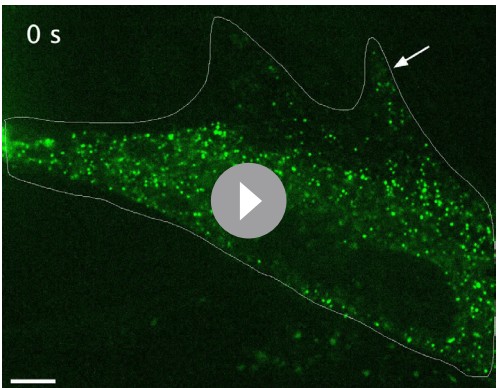

**Video 1.** Time-lapse imaging of *β-ACTIN-8xMS2* mRNA in live HeLa cells. The sfGFP foci of the *β-ACTIN-3′UTR-8xMS2* mRNA localize to the lamellipodia in HeLa cells. The white dashed line demarcates the cell. The white arrow indicates lamellipodia. Time interval, 2 s. Scale bar, 5 µm.
https://elifesciences.org/articles/82178/figures#video1

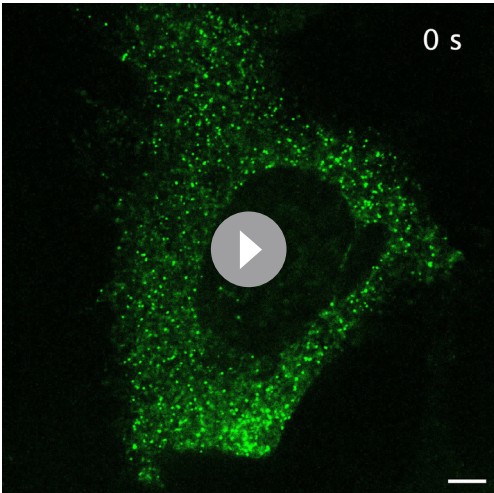

**Video 2.** Time-lapse imaging of sfGFP foci of *β-ACTIN-8xMS2* mRNA with a time interval of 1 s in HeLa cells. Scale bar, 5 μm.

https://elifesciences.org/articles/82178/figures#video2

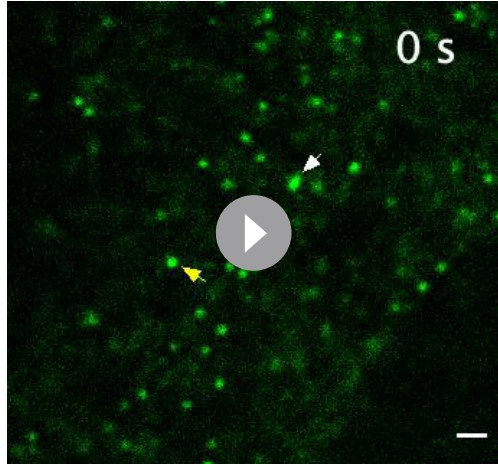

**Video 4.** Time-lapse imaging showing fission events of sfGFP foci of *β-ACTIN-8xMS2* mRNA in HeLa cells where large sfGFP foci split into smaller spots (white and yellow arrows). Time interval, 1 s. Scale bar, 1 μm.

https://elifesciences.org/articles/82178/figures#video4

smaller velocity (*Figure 1—figure supplement 5A, B*, *Figure 1—source data 1*), indicating that heavier labeling affected mRNA movement speed. In contrast, mRNAs labeled by MASS with MCP-6xSuntag showed a similar velocity to that labeled with the conventional 24xMS2 system (*Figure 1—figure supplement 5A, B*, *Figure 1—source data 1*). Those data pointed out that when MASS is used to measure the speed of mRNA movement, a short Suntag array (MCP-6xSuntag) should be used. We next measured the average intensity of each GFP foci of *β-ACTIN* mRNA labeled using the conventional 24xMS2 system or MASS with different lengths of Suntag arrays (MCP-24xSuntag, MCP-12xSuntag, and MCP-6xSuntag). We found GFP foci detected by MASS showed higher intensity than those detected by the 24xMS2 system (*Figure 1—figure supplement 6*, *Figure 1—source data 1*). These data support that with MASS, a single mRNA molecule could be tethered with more GFP molecules, forming a GFP spot of higher fluorescence intensity. Such an application, retains the ability to image mRNA using low-power lasers, thus lowering any unwanted phototoxicity and photobleaching and still allowing the tracking of mRNA dynamics in high temporal resolution.

We then performed time-lapse imaging of the *β-ACTIN-3'UTR-8XMS2* mRNA with a time

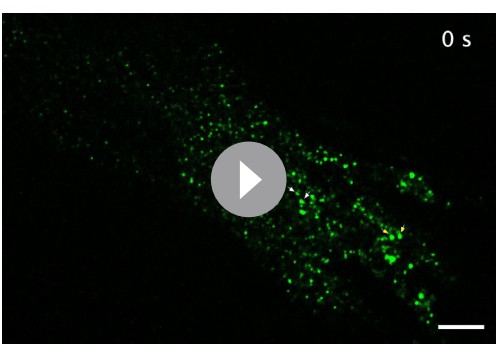

**Video 3.** Time-lapse imaging showing fusion events of sfGFP foci of *β-ACTIN-8xMS2* mRNA in HeLa cells where small sfGFP spots (white and yellow arrows) fuse into a single more prominent spot. Time interval, 2 s. Scale bar, 5μm.

https://elifesciences.org/articles/82178/figures#video3

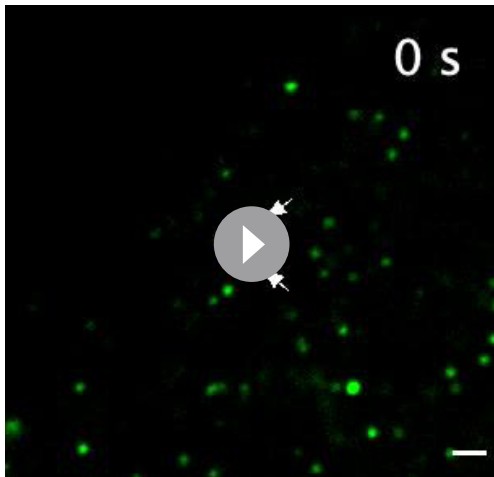

**Video 5.** Time-lapse imaging showing transient interactions of an sfGFP spot (yellow arrow) between two spots (white arrows) of *β-ACTIN-8xMS2* mRNA in HeLa cells. Time interval, 2 s. Scale bar, 1 μm.

https://elifesciences.org/articles/82178/figures#video5

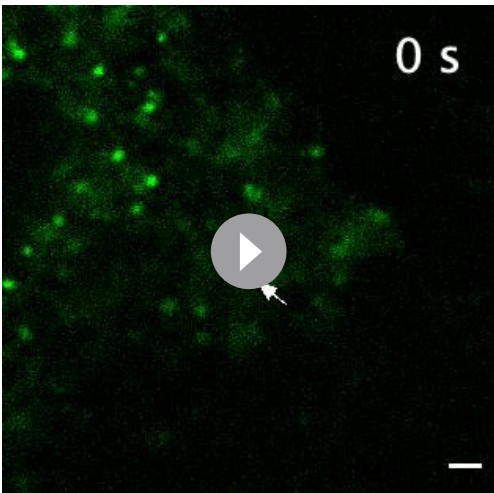

**Video 6.** Time-lapse imaging in HeLa cells showing an sfGFP spot of *β-ACTIN-8xMS2* mRNA showing no movement over a 10-s period. Time interval, 1 s. Scale bar, 1 µm.

https://elifesciences.org/articles/82178/figures#video6

interval of 1 s in HeLa cells (*Video 2*). We found that foci of mRNAs showed various dynamics: (1) Fusion. GFP spots fused into a more prominent spot (*Figure 1E* and *Video 3*); (2) Fission. Large GFP foci split into smaller spots (*Figure 1F* and *Video 4*); (3) Transient interaction. GFP foci touched each other briefly, then moved away (*Figure 1G* and *Video 5*), suggesting there are dynamic RNA–RNA interactions in cells; (4) Dynamic movement or anchoring. Despite most foci of *β-ACTIN-3'UTR-8XMS2* mRNAs showing dynamic movement in cells, other foci were far more static, showing little movement (*Figure 1H* and *Video 6*), suggesting that these latter mRNAs may be anchored to subcellular structures.

Our ultimate goal was to develop tools for endogenous mRNA imaging in live animals. It has been previously reported that the knock-in of short sequences into the genome is far more efficient than those of longer sequences (*Wang et al., 2022*; *Paix et al., 2017*). The MASS exploits this advantage as only 8xMS2 (350 nt) needs to be inserted into a genomic locus, thus overcoming the previous obstacle of the requirement of inserting a long 1300 nt 24xMS2 into the genome for live-cell imaging of endogenous mRNA.

We then used the nematode *C. elegans* to specifically examine whether the MASS could be used for RNA imaging in live animals. An 8xMS2 was placed into the 3'UTR of *cdc-42* mRNA (*Figure 2—figure supplement 1*). *cdc-42-8xMS2*, MCP-24xSuntag, and scFv-sfGFP were also expressed in the epidermis of live *C. elegans*. Consistent with the observations in HeLa cells, bright GFP foci could only be detected when all the required elements were present (*Figure 2—figure supplement 1*). Similarly, excluding any essential elements resulted in a complete failure of foci formation (*Figure 2—figure supplement 1*). Thus, the MASS was efficient in imaging exogenous RNAs in live animals.

Next, we set out to visualize gene expression activation and the dynamics of endogenous mRNAs in live animals. We used the skin of *C. elegans* as a model, which is composed of an epidermal epithelium with multiple nuclei. Upon wounding the epithelium via laser or needle, specific gene expressions and downstream signaling cascades for wound repair are then triggered and activated (*Xu and Chisholm, 2014*; *Figure 2A*). To this end, 8xMS2 was knocked into the 3'UTR region of two endogenous genes, *C42D4.3* and *mai-1* (*Figure 2B* and *Figure 2—figure supplement 2A*), the expression levels of which were reported to increase significantly after wounding (*Fu et al., 2020*). MCP-24xSuntag and scFv-sfGFP were expressed in the epidermis with the tissue-specific promoter *semo-1* and *col-19*. We then used a UV laser to injure an area of the epidermis. Prior to wounding, few sfGFP foci were detected in either wild-type (WT) or in 8xMS2 knock-in animals (*Figure 2B* and *Figure 2—figure supplement 2A*). In contrast, numerous GFP foci were formed 15 min after wounding in the *C42D4.3* and *mai-1* 8xMS2 knock-in animals but not in the WT animals (*Figure 2B* and *Figure 2—figure supplement 2A*, *Videos 7–9*). This result suggests that the wounding had activated *C42D4.3* and *mai-1* mRNA expression. In agreement with this, qRT-PCR showed that *C42D4.3* and *mai-1* mRNA levels were upregulated more than eightfold 15 min after injury (*Figure 2C* and *Figure 2—figure supplement 2B*, *Figure 2—source data 1*), confirming that GFP foci were able to detect endogenous *C42D4.3–8xMS2* and *mai-1-8xMS2* mRNAs. In addition, the expression level (*Figure 2—figure supplement 2C*, *Figure 2—source data 1*) and stability of *C42D4.3* mRNA (*Figure 2—figure supplement 2D*, *Figure 2—source data 1*) were similar in WT, *C42D4.3* 8xMS2 knock-in animals and animals expressing the MASS imaging system, indicating MASS did not affect the expression level and stability of endogenous *C42D4.3* mRNAs.

As the mRNA expression level of *C42D4.3* and *mai-1* significantly increased after wounding, we expected a boost in transcription. GFP will form extensive foci in the nuclei with active transcription.

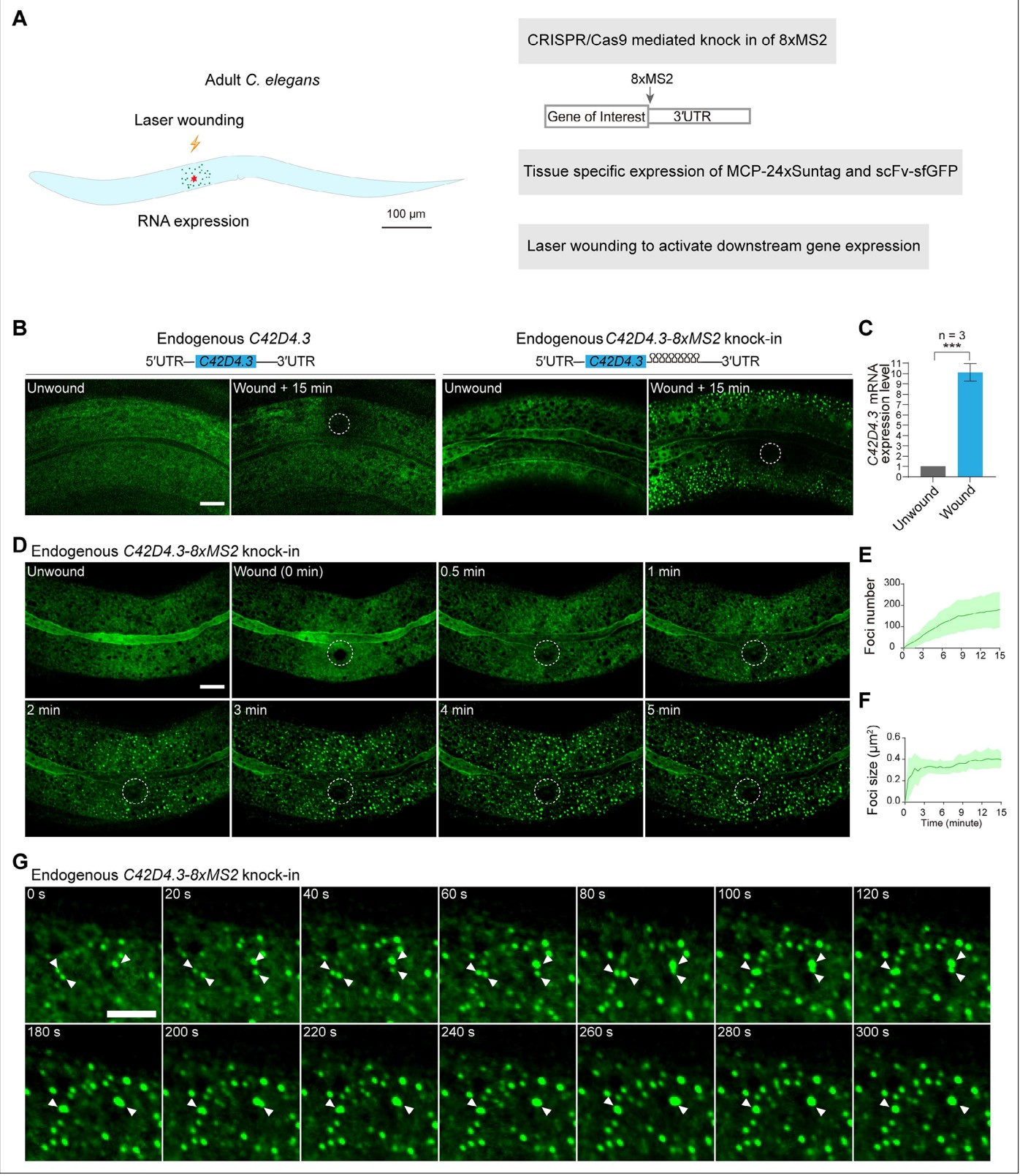

**Figure 2.** Live imaging of endogenous mRNAs in the epidermis of *C. elegans* using the MS2-based signal amplification with the Suntag system. (**A**) Schematic of the strategy for live imaging of endogenous mRNAs in the epidermis of *C. elegans*. (**B**) Representative images of endogenous *C42D4.3-8xMS2* mRNA in the epidermis of live *C. elegans* using the strategy described in (**A**). Left: *C42D4.3* without 8xMS2. Right: *C42D4.3* with 8xMS2. Images were taken before and 15 min after wounding. White dashed circles indicate wound sites. Scale bar, 10 μm. (**C**) Quantitative RT-PCR showing

*Figure 2 continued on next page*

*Figure 2 continued*

the expression level of endogenous *C42D4.3* mRNA in *C. elegans* before and 15 min after wounding. *n* = 3 independent experiments; bars indicate mean ± standard deviation (SD). Mann–Whitney test, ***p < 0.001. See also *Figure 2—source data 1*. (**D**) Time-lapse imaging of endogenous *C42D4.3-8xMS2* mRNA in the epidermis of live *C. elegans* before and after wounding. White dashed circles indicate the wound sites. Scale bar, 10 μm. Shown are mean ± SD of quantification of the number (**E**) and size (**F**) of sfGFP foci (endogenous *C42D4.3* mRNA) formed in the epidermis as measured 15 min after wounding. *n* = 10. (**G**) Time-lapse imaging showing fusion of endogenous *C42D4.3* foci (white arrows) after laser wounding. Scale bar, 5 μm.

The online version of this article includes the following source data and figure supplement(s) for figure 2:

**Source data 1.** qRT-PCR (Quantitative Reverse Transcription PCR), number, and size of foci of mRNAs detected by MASS in *C. elegans*.

**Figure supplement 1.** Live imaging of *cdc42* mRNA in *C. elegans* using MS2-based signal Amplification with Suntag System.

**Figure supplement 2.** Live imaging of endogenous mRNA in the epidermis of *C. elegans* using the MS2-based signal amplification with the Suntag system.

**Figure supplement 3.** Fast activation and spreading of endogenous gene expression in the epidermis of *C. elegans*.

**Figure supplement 4.** Treatment of Actinomycin D blocks the formation of *C42D4.3* mRNA foci.

**Figure supplement 5.** Foci of endogenous *C42D4.3-8xMS2* mRNAs showed a larger size than that of *BFP-8xMS2* mRNAs.

However, we failed to detect the appearance of bigger GFP foci in the nucleus. The epidermis of *C. elegans* is a syncytium with 139 nuclei located in different focal planes. With our microscopy, we could image only one focal plane, in which there are usually 4–10 nuclei. Therefore, it is likely that the nuclei with active transcription were out of focus, and therefore the GFP foci formed at the transcription site were not detected.

Next, we tracked the dynamics of endogenous *C42D4.3–8xMS2* mRNA in the *C. elegans* epidermis after wounding. We found that in proximity to the injury site GFP foci were detected as early as 1 min after wounding (*Figure 2D*, *Figure 2—figure supplement 3*, and *Videos 7 and 8*). As a control, we pretreated the *C. elegans* with Actinomycin D, which potently inhibits gene transcription. In such cases, no GFP foci could be detected after wounding (*Figure 2—figure supplement 4*). This indicated that GFP foci are newly synthesized *C42D4.3–8xMS2* mRNAs. Our data demonstrated that gene expression activation and transcription occur extremely fast (in this case, within 1 min after the stimulation). In addition, the appearance of GFP foci gradually spreads from the area around the injury site to distal regions. The total foci number steadily increased in the epidermis (*Figure 2D, E*, *Figure 2—figure supplement 3*, and *Videos 7 and 8*). These data suggested that wounding generated a signal to activate downstream gene expression around the injury site. The signal was then diffused to distal regions and able to induce gene expression there. In addition, GFP foci preferentially underwent fusion leading to increased foci size (*Figure 2F, G*, and *Video 10*, *Figure 2—source data 1*). In contrast, when exogenous *BFP-8xMS2* was expressed and imaged with MASS in the epidermis (*Figure 2—figure supplement 5A*, *Figure 2—source data 1*), such fusion events were not frequently observed (*Video 11*), and the puncta size of *BFP-8xMS2* mRNAs kept constant in 5 min (*Figure 2—figure supplement 5A–C*, *Figure 2—source data 1*). In addition, the puncta size of *C42D4.3-8xMS2*

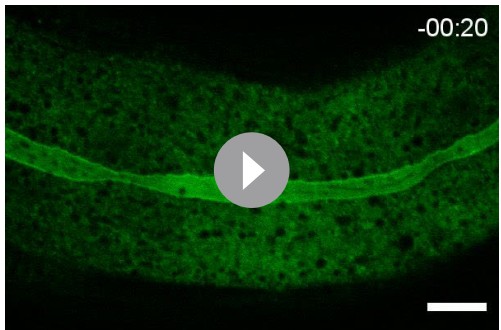

**Video 7.** Time-lapse imaging of endogenous *C42D4.3-8xMS2* mRNA dynamics in the epidermis of *C. elegans* after laser wounding. Time interval, 5 s. Scale bar, 10 μm.
https://elifesciences.org/articles/82178/figures#video7

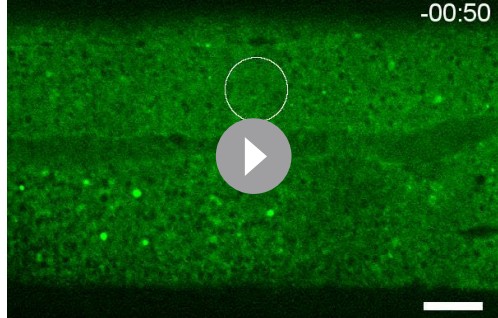

**Video 8.** Time-lapse imaging of endogenous *C42D4.3-8xMS2* mRNA dynamics in the epidermis of *C. elegans* after laser wounding. A different worm was shown. Time interval, 5 s. Scale bar, 10 μm.
https://elifesciences.org/articles/82178/figures#video8

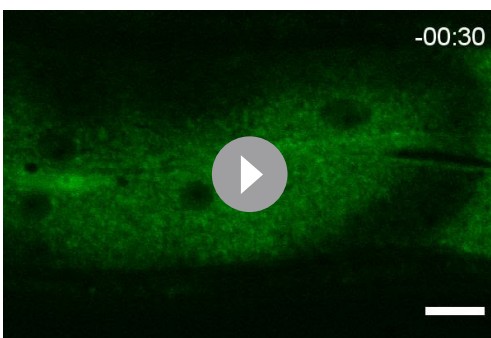

**Video 9.** Time-lapse imaging of endogenous *mai-1-8xMS2* mRNA dynamics in the epidermis of *C. elegans* after laser wounding. Time interval, 2 s. Scale bar, 10 μm.

https://elifesciences.org/articles/82178/figures#video9

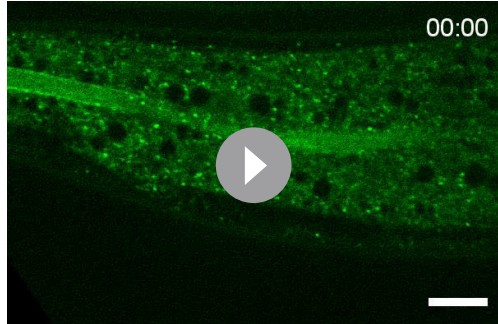

**Video 11.** Time-lapse imaging of exogenous *BFP-8xMS2* mRNA dynamics in the epidermis of *C. elegans*. Time interval, 0.5 s. Scale bar, 10 μm.

https://elifesciences.org/articles/82178/figures#video11

mRNAs was significantly larger than that of *BFP-8xMS2* mRNAs (*Figure 2—figure supplement 5C*, *Figure 2—source data 1*). These data suggested that *C42D4.3* mRNAs undergo clustering after wounding and form RNA granules in vivo. In agreement with our observation, it has been previously reported that mRNAs formed large clusters and are co-translated in *Drosophila* embryos (*Dufourt et al., 2021*).

## Discussion

It has been recently reported that the SunRISER and MoonRISER system by combination of PP7 and Suntag or Moontag enables imaging of single exogenous mRNAs in living cells. Through computational and experimental approaches, the authors further optimized the SunRISER system and showed that SunRISER provided an excellent approach for long-term imaging of overexpressed mRNA in living cells (*Guo and Lee, 2022*). The principle of the SunRISER system and MASS is similar, which uses a protein signal amplifier to generate a higher fluorescence signal for RNA imaging. In this study, we compared MASS to the conventional 24xMS2 mRNA imaging system and characterized the effect of MASS on mRNA stability and dynamics. In addition, we primarily explored the application of MASS to image endogenous RNA in live animals, which has not been tested in the previous study. Here, we showed such signal amplification strategy is valuable for imaging endogenous mRNAs that utilizing only 8xMS2 in comparison to the 24xMS2 used in the classic MS2-based live-cell RNA imaging. The advantage of a short MS2 is of prime benefits for imaging endogenous mRNA as it reduced difficulties involved in inserting a long 1300 nt 24xMS2 into a genomic locus. We expect this tool will help promote studies of RNA transcription, nuclear export, subcellular localization, translation, RNA sensing, and degradation, at the endogenous level in culture cells and live animals.

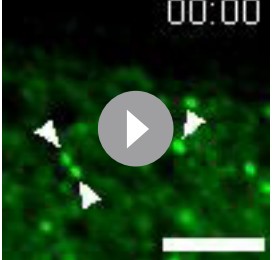

**Video 10.** Time-lapse imaging showing fusion events of sfGFP foci of endogenous *C42D4.3-8xMS2* mRNA in the epidermis of *C. elegans* where small sfGFP spots (white arrows) fused into a more prominent spot. Time interval, 5 s. Scale bar, 5 μm.

https://elifesciences.org/articles/82178/figures#video10

## Materials and methods
### Cell lines

The HEK293T/17 and NIH/3T3 cell lines were purchased from Procell. The human cervical cancer cell line, HeLa, was a gift from the lab of Christine Mayr (Memorial Sloan Kettering Cancer Center). Cells were maintained at 37°C with 5% $CO_2$ in Dulbecco's Modified Eagle Medium (DMEM) containing 4500 mg/l glucose, 10% fetal bovine serum, 100 U/ml penicillin, and 100 mg/ml streptomycin.

## Worm culture

All strains were cultured on the nematode growth medium (NGM) plates with *E. coli* OP50 at 20–22.5°C, unless otherwise indicated. The N2 Bristol strain was used as the WT strain.

## Constructs for mammalian cells

All PCR reactions were performed using KOD One PCR Master Mix -Blue- (TOYOBO). Recombinational cloning was performed with the ClonExpress One Step Cloning Kit (Vazyme).

## MS2 constructs

For MS2 V1 constructs, 2xMS2 was designed based on the MS2 sequence as reported by the Singer Lab (Addgene, #27118) (*Bertrand et al., 1998*). The sequence of 2xMS2 was as follows: ctgcaggtcgac tctagaaaacatgaggatcacccatgtctgcaggtcgactctagaaaacatgaggatcacccatgt. The EcoRI-2xMS2-EcoRV was synthesized and inserted into a pcDNA-puro-BFP backbone with EcoRI and EcoRV restriction sites to make the pcDNA-puro-BFP-2xMS2 construct. The EcoRV-2xMS2-XhoI was synthesized and inserted into the pcDNA-puro-BFP-2xMS2 backbone with EcoRV and XhoI restriction sites to make the pcDNA-puro-BFP-4xMS2 construct. The XhoI-2xMS2-ApaI was synthesized and inserted into the pcDNA-puro-BFP-4xMS2 backbone with XhoI and ApaI restriction sites to make the pcDNA-puro-BFP-6xMS2 construct. The BamHI-2xMS2-EcoRI was synthesized and inserted into the pcDNA-puro-BFP-6xMS2 backbone with BamHI and EcoRI restriction sites to make the pcDNA-puro-BFP-8xMS2 construct. Ligation was performed with T4 DNA ligase.

To make pcDNA-puro-BFP-*β-ACTIN*-3′UTR-8xMS2, the *β-ACTIN* coding sequence with a 3′UTR of 373 bp was PCR amplified from the cDNA of HEK293T/17 cells and inserted into the pcDNA-puro-BFP-8xMS2 vector with KpnI and BamHI restriction sites through recombinational cloning. The primers were *β-ACTIN* F and *β-ACTIN* R. The *β-ACTIN* coding sequence with a 3′UTR of 373 bp was *β-ACTIN* F: TGAATCTGTACAAGAAGCTTGGTACCGATGATGATATCGCCGCGCTCG, *β-ACTIN* R: TTCCACCA CACTGGACTAGTGGATCCAAGCAATGCTATCACCTCCCCTG. This fragment was also inserted into the pcDNA-puro-BFP-6xMS2 vector with KpnI and BamHI restriction sites through recombinational cloning to get pcDNA-puro-BFP-*β-ACTIN*-3′UTR-6xMS2.

To make pcDNA-puro-BFP-*β-ACTIN*-3′UTR, a *β-ACTIN* coding sequence with a 3′UTR of 373 bp was PCR amplified from the cDNA of HEK293T/17 cells and inserted into the pcDNA-puro-BFP vector with KpnI and BamHI sites through recombinational cloning. The primers were *β-ACTIN* F and *β-ACTIN* R.

To make the pcDNA-puro-BFP-*β-ACTIN*-3′UTR-24xMS2 V1, the BFP-*β-ACTIN*-3′UTR sequence was cut off from the pcDNA-puro-BFP-*β-ACTIN*-3′UTR-6xMS2 V1 vector and inserted into the pcDNA-puro-EGFP-24xMS2 (EGFP was cleaved off with the same enzymes) vector with NheI and EcoRI sites through ligation.

To make the pcDNA-puro-BFP-*β-ACTIN*-3′UTR-8xMS2 V7, 8xMS2 V7 sequence was designed based on the Addgene construct (#140705) (*Tutucci et al., 2018*). The BamHI-8xMS2 V7-EcoRI was synthesized and inserted into the pcDNA-puro-BFP-*β-ACTIN*-3′UTR vector with BamHI and EcoRI restriction sites. The sequence of 8xMS2 V7 was referred to the pET263-pUC57 24xMS2V7 (Addgene, #140705). The sequence we synthesized was as follows: ggatcctaaggtacctaattgcctagaaaggagcagacga tatggcgtcgctccctgcaggtcgactctagaaaccagcagagcatatgggctcgctggctgcagtattcccgggttcattagatcct aaggtacctaattgcctagaaaggagcagacgatatggcgtcgctccctgcaggtcgactctagaaaccagcagagcatatgggctcg ctggctgcagtattcccgggttcattagatcctaaggtacctaattgcctagaaaggagcagacgatatggcgtcgctccctgcaggt cgactctagaaaccagcagagcatatgggctcgctggctgcagtattcccgggttcattagatcctaaggtacctaattgcctagaaa ggagcagacgatatggcgtcgctccctgcaggtcgactctagaaaccagcagagcatatgggctcgctggctgcagtattcccgggtt cattagatccgaattc.

To make pcDNA-puro-BFP-*C-MYC*-3′UTR, the *C-MYC*-3′UTR with a length of 250 bp was PCR amplified from cDNA of HeLa cells with primers *C-MYC*-3′UTR F and *C-MYC*-3′UTR R. The *C-MYC*-3′UTR was inserted into the pcDNA3.1-puro-BFP vector with KpnI and BamHI restriction sites through recombinational cloning. *C-MYC*-3′UTR F: TGAATCTGTACAAGAAGCTTGGTACCCCCCTCAACG TTAGCTTCACC, *C-MYC*-3′UTR R: CTGCAGGGATCCGTAAATCTTAAAATTTTTTAAAAACAATTCTT AAATACAAATCTGTT.

To make pcDNA-puro-BFP-*C-MYC*-3′UTR-8xMS2, the *C-MYC*-3′UTR with a length of 250 bp was PCR amplified from cDNA of HeLa cells with primers *C-MYC*-3′UTR F and *C-MYC*-3′UTR R. The *C-MYC*-3′UTR was inserted into the pcDNA3.1-puro-BFP-8xMS2 vector with KpnI and BamHI restriction sites

through recombinational cloning. *C-MYC*-3′UTR F: TGAATCTGTACAAGAAGCTTGGTACCCCCCTC AACGTTAGCTTCACC, *C-MYC*-3′UTR R: CTGCAGGGATCCGTAAATCTTAAAATTTTTTAAAAACAAT TCTTAAATACAAATCTGTT.

To make pcDNA-puro-BFP-*HSPA1A*-3′UTR, the *HSPA1A*-3′UTR with a length of 200 bp was PCR amplified from cDNA of HeLa cells with primers *HSPA1A*-3′UTR F and *HSPA1A*-3′UTR R. The *C-MYC*-3′UTR was inserted into the pcDNA3.1-puro-BFP vector with KpnI and BamHI restriction sites through recombinational cloning. *HSPA1A*-3′UTR F: AAGCTTGGTACCGCCAAAGCCGCGGCGATC, *HSPA1A*-3′UTR R: CTGCAGGGATCCGTATTAAAAAGAAGAAATAGTCGTAAGATGGCAGTATAAATTCA.

To make pcDNA-puro-BFP-*HSPA1A*-3′UTR-8xMS2, the *HSPA1A*-3′UTR with a length of 200 bp was PCR amplified from cDNA of HeLa cells with primers *HSPA1A*-3′UTR F and *HSPA1A*-3′UTR R. The *C-MYC*-3′UTR was inserted into the pcDNA3.1-puro-BFP-8xMS2 vector with KpnI and BamHI restriction sites through recombinational cloning. *HSPA1A*-3′UTR F: AAGCTTGGTACCGCCAAAGCCGCG GCGATC, *HSPA1A*-3′UTR R: CTGCAGGGATCCGTATTAAAAAGAAGAAATAGTCGTAAGATGGC AGTATAAATTCA.

To make pcDNA-puro-BFP-*KIF18B*-3′UTR, the *KIF18B*-3′UTR with a length of 1131 bp was PCR amplified from cDNA of HeLa cells with primers *KIF18B*-3′UTR F and *KIF18B*-3′UTR R. The *KIF18B*-3′UTR was inserted into the pcDNA3.1-puro-BFP vector with KpnI and BamHI restriction sites through recombinational cloning. *KIF18B*-3′UTR F: TGAATCTGTACAAGAAGCTTGGTACCGCAGTGGAGG ACAGCACG, *KIF18B*-3′UTR R: CTGCAGGGATCCATCTTCACCAGGACTGTGGTTGG.

To make pcDNA-puro-BFP-*KIF18B*-3′UTR-8xMS2, the *KIF18B*-3′UTR with a length of 1131 bp was PCR amplified from cDNA of HeLa cells with primers *KIF18B*-3′UTR F *KIF18B*-3′UTR R. The *KIF18B*-3′UTR was inserted into the pcDNA3.1-puro-BFP-8xMS2 vector with KpnI and BamHI restriction sites through recombinational cloning. *KIF18B*-3′UTR F: TGAATCTGTACAAGAAGCTTGGTACCGCAGTG GAGGACAGCACG, *KIF18B*-3′UTR R: CTGCAGGGATCCATCTTCACCAGGACTGTGGTTGG.

## Suntag and MCP constructs

To make the pcDNA-MCP-mCherry-6xGCN4 construct, the 6xSuntag (6xGCN4) was PCR-amplified from pcDNA4TO-24xGCN4_v4-kif18b-24xPP7 (Addgene, #74928) (*Yan et al., 2016*) with primers 6xSuntag F and 6xSuntag R. The 6xSuntag was inserted into the pcDNA-MCP-mCherry vector with BsrGI and ApaI restriction sites through recombinational cloning. 6xSuntag F: CACCGGCGGCAT GGACGAGCTGTACAAGGGTGGAGGTTCTGGAGGA, 6xSuntag R: GCTGATCAGCGGGTTTAAAC GGGCCCTTATCCTGAGCCGGAACC.

To make the pcDNA3.1-tdMCP-6xGCN4, the tdMCP was cut off from the pcDNA-puro-tdMCP-12xSuntag and inserted into the pcDNA-MCP-mCherry-6xGCN4 vector with KpnI and BsrGI restriction sites through ligation.

To make the pcDNA-puro-MCP-mCherry-12xSuntag construct, the 12xSuntag (12xGCN4) was PCR amplified from pcDNA4TO-24xGCN4_v4-kif18b-24xPP7 (Addgene, #74928) (*Yan et al., 2016*) with primers 12xSuntag F and 12xSuntag R. The 12xSuntag was inserted into the pcDNA-MCP-mCherry vector with BsrGI and ApaI restriction sites through recombinational cloning. 12xSuntag F: CACC GGCGGCATGGACGAGCTGTACAAGGGTGGAGGTTCTGGAGGA, 12xSuntag R: GCTGATCAGCGG GTTTAAACGGGCCCTTAGCCCGAGCCCGAGCC.

To make the pcDNA3.1-mCherry-12xGCN4, the mCherry-12xGCN4 was PCR amplified from the pcDNA-puro-MCP-mCherry-12xSuntag with primers mCherry-12xSuntag F and mCherry-12xSuntag R. The mCherry-12xSuntag was inserted into the pcDNA-MCP-mCherry vector with KpnI and XbaI restriction sites through recombinational cloning. mCherry-12xSuntag F: AGCGTTTAAACTTAAG GCTTGGTACCATGGTGAGCAAGGGCGAGGAGGA, mCherry-12xSuntag R: GCTGATCAGCGG GTTTAAACGGGCCCTTAGCCCGAGCCCGAGCC.

To make the pcDNA3.1-tdMCP-12xGCN4, the tdMCP was PCR amplified from the phage UBC NLS-HA-2XMCP-tagRFPt (Addgene, #64541) (*Halstead et al., 2015*) with primers tdMCP-12xSuntag F and tdMCP-12xSuntag R. The tdMCP was inserted into the pcDNA-mCherry-12xSuntag vector with KpnI and BsrGI restriction sites through recombinational cloning. tdMCP-12xSuntag F:AGCGTTTA AACTTAAGGCTTGGTACCTGCTAGCCGTTAAAATGGCTTCTAAC, tdMCP-12xSuntag R: TCCTCCAG AACCTCCACCCTTGTACATCACCATTCTAGAATCCGCGTAGAT.

To make the pcDNA-puro-MCP-mCherry-24xSuntag construct, the 24xSuntag (24xGCN4) was PCR amplified from pcDNA4TO-24xGCN4_v4-kif18b-24xPP7 (Addgene, #74928) with primers 24xSuntag

F and 24xSuntag R. The 24xSuntag was inserted into the pcDNA-MCP-mCherry vector with BsrGI and ApaI restriction sites through recombinational cloning. 24xSuntag F: CACCGGCGGCATGGACGAGC TGTACAAGGGTGGAGGTTCTGGAGGA, 24xSuntag R: GCTGATCAGCGGGTTTAAACGGGCCCTT AACCCGAGCCAGAACC.

To make the pcDNA3.1-mCherry-24xGCN4, the mCherry-24xGCN4 was PCR amplified from the pcDNA-puro-MCP-mCherry-24xSuntag with primers mCherry-24xSuntag F and mCherry-24xSuntag R. The mCherry-24xSuntag was inserted into the pcDNA-MCP-mCherry vector with KpnI and XbaI restriction sites through recombinational cloning. mCherry-24xSuntag F: AGCGTTTAAACTTAAG GCTTGGTACCATGGTGAGCAAGGGCGAGGAGGA, mCherry-24xSuntag R: GCTGATCAGCGG GTTTAAACGGGCCCTTAACCCGAGCCAGAACC.

To make the pcDNA3.1-tdMCP-24xGCN4, the tdMCP was cut off from the pcDNA-puro-tdMCP-12xSuntag and inserted into the pcDNA3.1-mCherry-24xGCN4 vector with KpnI and BsrGI sites through ligation.

To make the pcDNA-puro-mCherry-24xSuntag construct, the mCherry-24xSuntag was PCR amplified from the pcDNA-puro-MCP-mCherry-24xSuntag vector with primers mCherry-24xSuntag F and mCherry-24xSuntag R. The MCP-mCherry was then cut off from the pcDNA-puro-MCP-mCherry backbone by KpnI and XbaI digestion and replaced with the mCherry-24xSuntag through recombinational cloning. mCherry-24xSuntag F: AGCGTTTAAACTTAAGGCTTGGTACCATGGTGAGCAAGGGCGAG GAGGA, mCherry-24xSuntag R: CAGCGGGTTTAAACGGGCCC.

To make the pcDNA-puro-scFv-sfGFP construct, the scFV-HAtag-sfGFP-GBI was PCR amplified from pHR-scFv-GCN4-sfGFP-GB1-dWPRE (Addgene, #60907) with primers scFV-HAtag-sfGFP-GBI F and scFV-HAtag-sfGFP-GBI R. The scFV-HAtag-sfGFP-GBI was inserted into the pcDNA-puro vector with NheI and HindIII restriction sites through recombinational cloning. scFV-HAtag-sfGFP-GBI F: ACCCAAGCTGGCTAGCAACCATGGGCCCCGACATCGTG, scFV-HAtag-sfGFP-GBI R: GTGGATCC GAGCTCGGTACCAAGCTTTTATTCGGTTACCGTGAAGGTTTTGGTA.

To make the pcDNA-puro-sfGFP construct, the HAtag-sfGFP-GBI was PCR amplified from the pcDNA-puro-scFv-sfGFP vector with primers HAtag-sfGFP-GBI F and HAtag-sfGFP-GBI R. The HAtag-sfGFP-GBI was inserted into the pcDNA-puro vector with NheI and HindIII restriction sites through recombinational cloning. HAtag-sfGFP-GBI F: CTATAGGGAGACCCAAGCTGGCTAGCAACCATGTAC CCATACGATGTTCCAGATTACG, HAtag-sfGFP-GBI R: GCACACCACACTGGACTAGTGG.

To make the pcDNA3.1-tdMCP-sfGFP, the sfGFP was PCR amplified from the pcDNA-puro-scFv-sfGFP with primers tdMCP-sfGFP F and tdMCP-sfGFP R. The sfGFP was inserted into the pcDNA-tdMCP-12xGCN4 vector with BsrGI and ApaI restriction sites through recombinational cloning. tdMCP-sfGFP F: CGCGGATTCTAGAATGGTGATGTACAGTTCAGGAGGCGGCGGAA, tdMCP-sfGFP R: GCTGATCAGCGGGTTTAAACGGGCCCTTATTCGGTTACCGTGAAGGTTTTGGTA.

To make the pcDNA3.1-NLS-tdMCP-sfGFP, the NLS sequence was synthesized and inserted into the pcDNA3.1-tdMCP-sfGFP vector with NheI restriction sites. The sequence of NLS is as follows: ccaaaaaagaaaagaaaagtt.

## Constructs for *C. elegans*

Recombinational cloning was performed with the ClonExpress One-Step Cloning Kit (Vazyme).

P*col-19*-mKate2::CDC-42–8xMS2 was generated through recombinational cloning of two PCR amplified fragments. Fragment 1: P*col-19*-mKate2::CDC-42 was PCR amplified from the P*col-19*-mKate2::CDC-42 construct with primers *zju4844*: CAGCTTTCTTGTACAAAGTGGTGATATCAAG and *zju4845*: CTAGAGAATATTGCACTTCTTCTTCTTCCTG. Fragment 2: 8xMS2 was PCR amplified from the pcDNA-puro-BFP-*β-ACTIN*-3'UTR-8xMS2 with primers *zju4555*: CAGGAGAAGAAGAAGA AGTGCAATATTCTCTAGCTGCAGGTCGACTCTAGAAAAC and *zju4556*: GCTGGGTCGAATTCGC CCTTACATGGGTGATCCTCATGTTTTC.

P*semo-1*-MCP-24xSuntag was generated through recombinational cloning of two PCR amplified fragments. Fragment 1: P*semo-1* was PCR-amplified from the P*semo-1*-tomm-20-linker-mKate2-tbb construct with primers *zju4540*: TGAGACTTTTTTCTTGGCGG and *zju4541*: AGCCTGCTTTTTTGTA CAAACTTG. Fragment 2: MCP-24xSuntag was PCR amplified from the pcDNA-puro-MCP-mCherry-24xSuntag construct with primers *zju4538*: TCACAAGTTTGTACAAAAAAGCAGGCTATGGCTTCT AACTTTACTCAGTTCG and *zju4539*: GTGCCGCCAAGAAAAAAGTCTCATTAACCCGAGCCAGAAC CC.

P*col-19*-BFP-8xMS2 was generated through recombinational cloning of two PCR amplified fragments. Fragment 1: P*col-19*-8xMS2 was PCR amplified from the P*col-19*-mKate2::CDC-42–8xMS2 with primers *zju4764*: GGGAGGTGATAGCATTGCTTGG and *zju4765*: TCAGCGGGTTTAAACGGG. Fragment 2: BFP was PCR amplified from the P*col-19*-lifact-BFP with primers *zju5441*: CCCGTTTA AACCCGCTGAATTTGCGTCGCTGCAATTCTTATCAC and *zju5442*: ATCCAAGCAATGCTATCACC TCCCAGATCCGGCTCCATTAAGCTTGTG.

P*col-19*-scFv-sfGFP was generated through recombinational cloning of two PCR amplified fragments. Fragment 1: P*col-19* was PCR amplified from the pCR8-P*col-19*-GFP-CDC-42 construct with primers *zju4550*: AAGGGCGAATTCGACCCAGC and *zju4551*: ACCGGTGAGCTCTACCTGTAC. Fragment 2: scFv-sfGFP was PCR amplified from the pcDNA-puro-scFv-sfGFP construct with primers *zju4552*: GGTACAGGTAGAGCTCACCGGTGCAACCATGGGCCCCGAC and *zju4553*: GCTGGGTC GAATTCGCCCTTTTACTCTAGACTCGAGCGGC.

P*col-19*-sfGFP was generated by removing the scFv from the P*col-19*-scFv-sfGFP plasmid using KLD Enzyme mix (NEB) with primers *zju5038*: TACCCATACGATGTTCCAGATTACG and *zju5039*: GGTTGCACCGGTGAGCTC.

P*semo-1*-MCP was generated by removing the 24xSuntag from the P*semo-1*-MCP-24xSuntag plasmid using KLD Enzyme mix (NEB) with primers *zju5040*: TGAGACTTTTTTCTTGGCGGCA and *zju5041*: TCCTCCAGAACCTCCACC.

P*semo-1*–24xSuntag was generated by removing MCP from the plasmid P*semo-1*-MCP-24xSuntag plasmid using KLD Enzyme mix (NEB) with primers *zju5042*: GGATCATCAGGTGCTGGATCCG and *zju5043*: AGCCTGCTTTTTTGTACAAACTTG.

Constructs will be deposited to Addgene.

## Transgenic worms

Different combinations of plasmids were used for generating extrachromosomal array transgenic worms. Briefly, 50 ng/µl plasmids of P*semo-1*-MCP-24xSuntag, 10 ng/µl plasmids of P*col-19*-antibody-sfGFP and 10 ng/µl co-injection marker P*ttx-3-RFP* were injected into N2 and knock-in animals. The extrachromosomal strains were as follows: SHX3314: P*col-19*-mKate2::CDC-42–8xMS2, P*semo-1*-MCP-24xSuntag, P*col-19*-scFv-sfGFP(*zjuEx2144*); SHX3749: P*col-19*-mKate2::CDC-42–8xMS2; P*semo-1*-MCP; P*col-19*-scFv-sfGFP(*zjuEx2031*); SHX3751: P*col-19*-mKate2::CDC-42–8xMS2; P*semo-1*-MCP-24xSuntag; P*col-19*-sfGFP(*zjuEx2033*); SHX3755: P*col-19*-mKate2::CDC-42–8xMS2; P*semo-1*–24xSuntag; P*col-19*-scFV-sfGFP(*zjuEx2037*); SHX3757: P*col-19*-mKate2::CDC-42; P*semo-1*-MCP-24xSuntag; P*col-19*-scFv-sfGFP(*zjuEx2144*); SHX3908: *C42D4.3*–8xMS2(*syb5468*); P*semo-1*-MCP-24xSuntag; P*col-19*-scFv-sfGFP(*zjuEx2144*); SHX3909: *mai-1*–8xMS2(*syb5458*); P*semo-1*-MCP-24xSuntag; and P*col-19*-scFv-sfGFP(*zjuEx2144*); SHX4319: P*col-19*-BFP-8xMS2(*zjuEx2429*).

## Transfections

Lipofectamine 2000 (Invitrogen) and Endofectin MAX (Genecopoeia) were used for all transfections.

## Live-cell imaging

For images in *Figure 1* and *Videos 1–6*:

HeLa cells were plated on 35-mm glass-bottom dishes (NEST, 801001) at a density of $0.13 \times 10^6$. The indicated constructs were transfected into HeLa cells. Twelve to twenty-four hours after transfection, cells were imaged using a spinning-disk confocal microscope with a ×60 objective (Nikon T2 Microscope; Apo ×60 oil; 1.4 NA) using the SoRa mode. Exposure time was 500 ms. For time-lapse imaging, the time interval was set to 1 or 2 s. Images were analyzed with FIJI (ImageJ).

For images shown in supplements:

HeLa cells were plated on 35-mm glass-bottom dishes (BGI, BGX-03520-100) at a cell number of $0.8 \times 10^6$ to grow for 12 hr. The indicated constructs were transfected into HeLa cells. Twelve hours after transfection, cells were imaged by a spinning-disk confocal microscope with a ×60 objective (Olympus SpinSR10 Microscope; Apo ×60 oil; 1.5 NA) using the SoRa mode. Different parameters were used:

Images shown in supplements: Exposure time was 500ms.

Time-lapse imaging for calculating the speed of mRNA movement: Exposure time was 200 ms, and the interval was set to 217 ms. Fifteen frames are recorded.

Z-stack imaging for calculating intensity and signal-to-noise: Exposure time was 500 ms. Stacks of 5 planes with a z-spacing of 0.5 µm were obtained by using range mode.

## Single-molecule FISH and image acquisition in cells

Two 3′ Cy3 fluorescently labeled DNA oligos (probe-1: catgggtgatcctcatgt, probe-2: ttctagagtcgacctgca) as smFISH probes against MS2 stem-loops and the linker regions were synthesized by Tsingke. HeLa cells were plated on 35-mm glass-bottom dishes (BGI, BGX-03520-100) at a density of $0.8 \times 10^6$ to grow for 12 hr and transfected with the indicated constructs. Twelve hours after transfection, cells were washed with phosphate-buffered saline (PBS), fixed with 2% formaldehyde at 37°C for 10 min, and washed twice each for 5 min with PBS. PBS was then discarded, and 2 ml 70% ethanol was added. The plates were kept at 4°C for 8 hr. The 70% ethanol was aspirated, 1 ml wash buffer was added (2× SSC (saline sodium citrate), 10% formamide in RNase-free water), and incubated at RT for 5 min. Hybridization mix was prepared by mixing 10% Dextran sulfate, 10% formamide, 2× SSC, 2 mM ribonucleoside vanadyl complex (NEB), 200 µg/ml yeast tRNA (Sigma, 10109495001), 10 nM probe-1, and 10 nM probe-2. To each plate, 800 µl hybridization mix was added and hybridized at 37°C for 16 hr. Fixed cells on plates were washed twice for 30 min (each time 15 min) with pre-warmed wash buffer (1 ml, 37°C) in the dark, followed by one quick wash with PBST, and kept in PBST for imaging within 3 hr. Images were captured using confocal ZEISS LSM 880 with Airyscan super-resolution mode.

## CRISPR-Cas9-mediated gene knock-in in *C. elegans*

*C424.3–8xMS2(syb5468)* and *mai-1–8xMS2(syb5458)* knock-in animals were generated using the CRISPR-Cas9 system. Briefly, two repair templates were cloned using the Gibson assembly technique within a pDD282 plasmid. The MS2 sequence was designed after the termination codon of two genes in the repair template. sgRNA, repair templates, and *Peft-3-Cas9-NLS-pU6-dpy-10* sgRNA, as well as *Pmyo-2-cherry* as co-injection markers, were injected into N2 worms. Knock-in animals were confirmed by PCR genotyping and sequencing. Roller or dumpy animals were heat-shocked or outcrossed to remove markers. The sequences of the sgRNAs used in this study are as follows: *C42D4.3* sg1 ( CCAAAACTTGCTTGCCAGAACTT); *C42D4.3* sg2 (CCAGAACTTTCGGACAATAATTG); *mai-1* sg1 ( ACAACATCAGCAACGACTGAAGG); *mai-1* sg2 (CGACTGAAGGAAATCGAGAAAGG). The precise sequence knock-in is described as follows: gggaggtgatagcattgcttggatccctgcaggtcgactctagaaaacatga ggatcacccatgtctgcaggtcgac     tctagaaaacatgaggatcacccatgtgaattcctgcaggtcgactctagaaaacatgagga tcacccatgtctgcaggtcgactctagaaaacatgaggatcacccatgtgatatcctgcaggtcgactctagaaaacatgaggatcac ccatgtctgcaggtcgactctagaaaacatgaggatcacccatgtctcgagctgcaggtcgactctagaaaacatgaggatcacccat gtctgcaggtcgactctagaaaacatgaggatcacccatgtgggcccgtttaaacccgctga.

## Drug treatment

The Actinomycin D stock solution was dissolved in DMSO (Dimethyl sulfoxide) and diluted with M9 to a working concentration of 30 µM. Young adult stage worms were incubated in 100 µl Actinomycin D (APExBIO; Catalog No. A4448) solution (containing *E. coli* OP50) using a 1.5-ml microcentrifuge tube at 20°C for 3 hr. The worms were then transferred to fresh NGM plates to dry before wounding and imaging.

## mRNA stability test in *C. elegans*

Worms (*n* = 200) of WT, *C42D4.3*-8xMS2 knock-in animals, animals expressing the MASS imaging system, and *C42D4.3*-8xMS2 knock-in animals expressing the MASS imaging system were incubated in 200 µl Actinomycin D solution (containing *E. coil* OP50) using the 1.5 ml tubes at 20°C for 0, 3, or 6 hr. The treated worms were used to extract RNA for qRT-PCR (Quantitative Reverse Transcription PCR) of *C42D4.3* expression.

## mRNA stability test in cells

Three exogenous genes (*C-MYC*, *HSPA1A*, *KIF18B*) were selected. For each gene, two plasmids are constructed: with or without 8xMS2. Three groups of transfection were performed: BFP-*gene*-3′UTR, BFP-*gene*-3′UTR-8xMS2, and BFP-*gene*-3′UTR-8xMS2 co-transfected with tdMCP-24xSuntag and scFV-sfGFP. HeLa cells were plated on 12-well plates at a density of $0.3 \times 10^6$ to grow for 24 hr and transfected with the indicated constructs. After 12 hr of transfection, 5 µg/ml Actinomycin D (Act D,

APExBIO, A4448) was added, and also the first samples with no treatment were collected. Cells with Act D treatment were separately collected after 3 and 6 hr. All samples were used to extract RNA for qRT-PCR (Quantitative Reverse Transcription PCR).

## Wounding assay

A Micropoint UV laser was used to wound the epidermis of young adult stage worms. Briefly, worms at the young adult stage were mounted to 4% agarose gel on a slide, narcotized with 12 µM Levamisole, and wounded using a Micropoint UV laser. The energy ranged from 65 to 70. The repetition rate was 10 Hz, which repeats five times.

## Live imaging in *C. elegans*

Worms were imaged on a spinning disk confocal microscope Nikon Eclipse Ti with Andor confocal scanning unit (×100, NA 1.46 objective). Z-section and time-lapse were set using Andor IQ software to capture the images. For wound-induced reactions, images were single confocal planes imaged every 2 s for 930 s, including 30 s before UV laser and 900 s after wounding. Camera EM gain was 80. Green fluorescence was visualized with a 488-nm laser, and red fluorescence was visualized with a 561-nm laser. The exposure time of the GFP channel was 280 ms and the exposure time of the RFP channel was 120ms. For normal conditions, images were single confocal planes imaged every 0.5 s for 600 s. Camera EM gain was 140. The exposure time of the GFP channel was 110 ms.

## Quantification of the size and number of sfGFP foci

The size and number of sfGFP were quantified using Fiji software (https://imagej.net/imagej-wiki-static/Fiji). The background intensity was set as the threshold, and the size and number of foci were calculated using the Analyze Particles command. The value of puncta size and number was statistically analyzed using GraphPad. For puncta size and number, we chose 600 × 400 pixels around the wound site for analysis. The size and number changes were quantified using mean with standard deviation (SD).

## qRT-PCR (Quantitative Reverse Transcription PCR) in *C. elegans*

Total RNAs were extracted from 100 young adult worms with TRIzol reagent (Invitrogen, Carlsbad, CA, USA), quantitated by spectrophotometry using a NanoDrop (Thermo, USA), and reverse transcribed using HiScriptIIIReverseTranscriptase (Vazyme, China). qRT-PCR was performed with *rbd-1* as the house-keeping gene using the SYBR Green Supermix (Vazyme). The following primers were used in this study.

*rbd-1*, forward (fwd) CACGGAACAGCAACTACGGA, reverse (rev) CGGCTTGTTTGCATCACCAA ; *C42D4.3*, fwd GCCAGACTCTTGCCTCTCAA, rev CACGCGGTGTGATCTTTTCC; *mai-1*, fwd CGGCTCAATCCGTGAAGC, rev TGTTGGCTTTGCGTCATATC.

## qRT-PCR (Quantitative Reverse Transcription PCR) in cells

Total RNAs were extracted with RNA isolater (Vazyme, R401-01), and reverse transcribed using HiScriptIIIReverseTranscriptase (Vazyme, R323-01). qRT-PCR was performed with *GAPDH* as the house-keeping gene using the ChamQ Universal SYBR qPCR Master Mix (Vazyme, Q711-03). Primers for qRT-PCR of genes were designed as primer F at BFP and primer R at genes. The following primers were used in this study.

BFP-*C-MYC* F: TACTGCGACCTCCCTAGCAA,
BFP-*C-MYC* R: TACTGCGACCTCCCTAGCAA,
BFP-*HSPA1A* F: ACTGCGACCTCCCTAGCAA,
BFP-*HSPA1A* R: TCTCCACCTTGCCGTGTTGG,
BFP-*KIF18B* F: ATACTGCGACCTCCCTAGCA,
BFP-*KIF18B* R: CGCACCCGTACCACTACTTG,
*GAPDH* F: GCGAGATCCCTCCAAAATCAA,
*GAPDH* R: GTTCACACCCATGACGAACAT.

### Statistical analysis in *C. elegans*

Statistical analyses were performed using GraphPad Prism 7 (La Jolla, CA). One-way analysis of variance (ANOVA for multiple comparisons), a non-parametric Mann–Whitney test was used for two comparisons. NS indicates not significant, *indicates $p < 0.05$, **indicates $p < 0.01$, *** indicates $p < 0.001$, **** indicates $p < 0.0001$. Unless elsewhere stated, bars represent means ± SD.

## Quantification and statistical analysis in cells

### smFISH analysis

To quantify mRNA numbers detected by smFISH and MASS, also the ratio of colocalization, Fiji Plugin (https://imagej.net/imagej-wiki-static/Fiji)-ComDet was used for spot detection in two-color channel images. Data were analyzed in Excel, and the scatter diagram was generated using GraphPad Prism9.

### The mRNA expression level by qRT-PCR

Statistical analyses were performed using GraphPad Prism 9. One-way ANOVA for multiple comparisons and non-paired *t*-tests were used for two comparisons. NS indicates not significant, *indicates $p < 0.05$, **indicates $p < 0.01$, *** indicates $p < 0.001$, **** indicates $p < 0.0001$. Unless elsewhere stated, bars represent means ± SD.

### Intensity and signal-to-noise

Single-plane images were used for analysis using the Fiji plugin Trackmate (*Tinevez et al., 2017*). Particle size was estimated at 0.3 μm and a Differences of Gaussian (DoG) filter was applied to detect all spots. The 'Simple LAP tracker' particle-linking algorithm was used and the linking max distance was 15 μm, the gap closing distance was 15 μm, and the gap closing max frame was 2. The intensity and signal-to-noise ratio were produced by TrackMate and served as source data for the generation of histogram graphs by SPSS, and curve diagrams by Excel.

### Velocity analysis

For the conventional 24xMS2 mRNA imaging system, an NLS was fused to MCP to localize NLS-MCP-GFP into the nucleus, which allows the detection of mRNAs in the cytoplasm with a high signal-to-noise ratio. However, mRNAs in the nucleus cannot be detected clearly. Therefore, when the velocity analysis was performed, signals in the nucleus were excluded and only mRNA foci in the cytoplasm were included for analysis.

Single-molecule tracking was performed in 2D using the Fiji plugin Trackmate (*Tinevez et al., 2017*). To improve the accuracy of spot detection, we cropped the cytoplasm into several parts with no overlapping. Single particles were segmented frame-by-frame with a time interval of 217 ms which was set before starting the TrackMate. And the parameters for tracking single spots were the same as described in 'Intensity and signal-to-noise'. The velocity of each mRNA foci was produced by TrackMate and served as source data for the generation of histogram graphs by SPSS, and curve diagrams by Excel.

## Acknowledgements

We thank all members of the Ma lab for helpful discussions. We thank Christine Mayr (Memorial Sloan Kettering Cancer Center) for providing HeLa cells, pcDNA vectors, and Suntag constructs. We thank Jian Zhang (Yunnan University) for critically reading the manuscript and suggestions. This work was funded by the National Key R&D Program of China (2021YFA1101002, 2021YFA1300302), the National Natural Science Foundation of China (31972891), and the Zhejiang Province Natural Science Foundation (2-2060203-21-001) to SX; and the start-up funding from the Life Sciences Institute, the Leading innovation and entrepreneurship team of Hangzhou (TD2020006), the young fellows of Zhejiang University (2021QN81027) to WM.

## Additional information

### Funding

| Funder | Grant reference number | Author |
|---|---|---|
| National Natural Science Foundation of China | 31972891 | Suhong Xu |
| Leading innovation and entrepreneurship | TD2020006 | Weirui Ma |
| Zhejiang University | 2021QN81027 | Weirui Ma |
| National Key R&D Program of China | 2021YFA1101002 | Suhong Xu |
| National Key R&D Program of China | 2021YFA1300302 | Suhong Xu |
| Zhejiang Province Natural Science Foundation | 2-2060203-21-001 | Suhong Xu |

The funders had no role in study design, data collection, and interpretation, or the decision to submit the work for publication.

### Author contributions

Yucen Hu, Jingxiu Xu, Formal analysis, Validation, Investigation, Visualization, Methodology, Writing - original draft, Project administration, Writing - review and editing; Erqing Gao, Formal analysis, Validation, Investigation, Visualization, Methodology; Xueyuan Fan, Bingcheng Ye, Methodology; Jieli Wei, Supervision, Methodology; Suhong Xu, Conceptualization, Formal analysis, Supervision, Funding acquisition, Investigation, Visualization, Methodology, Project administration, Writing - review and editing; Weirui Ma, Conceptualization, Formal analysis, Supervision, Funding acquisition, Investigation, Visualization, Methodology, Writing - original draft, Project administration, Writing - review and editing

### Author ORCIDs

Xueyuan Fan http://orcid.org/0000-0001-8297-1596
Suhong Xu http://orcid.org/0000-0002-4079-340X
Weirui Ma http://orcid.org/0000-0002-9193-7311

### Decision letter and Author response

Decision letter https://doi.org/10.7554/eLife.82178.sa1
Author response https://doi.org/10.7554/eLife.82178.sa2

---

## Additional files

### Supplementary files
• MDAR checklist

### Data availability

Videos 1–11 contain source data for figures. Figure 1—source data 1 contains source data for statistical analysis in cells. Figure 2—source data 1 contains source data for statistical analysis in *C. elegans*.

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
