## [Editor Report]

The authors have amplified the signal on MS2 so that a smaller insertion is sufficient to track mRNA in vivo. They provide solid evidence that this approach generates a sufficient signal, equivalent to the full-length MS2. This work, along with a previously reported similar method will be useful to investigators considering single-molecule imaging in Elegans as well as other organisms.

---

## [Decision Letter]

**Decision letter after peer review:**

Thank you for submitting your article "Enhanced Single RNA Imaging Reveals Dynamic Gene Expression in Live Animals" for consideration by *eLife*. Your article has been reviewed by 4 peer reviewers, one of whom is a member of our Board of Reviewing Editors, and the evaluation has been overseen by James Manley as the Senior Editor. The reviewers have opted to remain anonymous.

The reviewers have discussed their reviews with one another, and the Reviewing Editor has drafted this to help you prepare a revised submission. All reviewers are positive about the concept of labeling by MASS but are not completely convinced that it is working without any effect on the mRNA. They all request further data that rigorously report on the mRNA tagged compared to non-tagged mRNA, or singly tagged mRNA.

Essential revisions:

The authors have not compared their MASS system to a standard in the field, the MS2 system. There are questions on the relevance of this approach for single molecule imaging, and the effect of the tagging on the physiology of the mRNA. There needs to be a more rigorous approach showing that the mRNA tagged in this way behaves as if it was not tagged. Suggestions also include comparison to FISH images in fixed cells, and measurement of velocities with and without the extensive tagging. It may be that the massive amount of protein on the mRNA may render it inert.

The referencing does not include similar work in the field (eg Guo et al) that needs to be discussed and compared more thoroughly.

Provide quantitative evaluation of signal and increase in signal to noise. Refer to the reviewers' many comments and suggestions.

*Reviewer #2 (Recommendations for the authors):*

1. Line 44: There are more previous works reporting endogenous mRNA imaging in live animals.

Forrest and Gavis, "Live imaging of endogenous RNA reveals a diffusion and entrapment mechanism for nanos mRNA localization in *Drosophila*", Current Biology, 13, 1159 (2003)

Lee et al., "Real-time visualization of mRNA synthesis during memory formation in live mice" PNAS, 119, e2117076119 (2022)

2. Line 410: More details need to be described for live imaging in *C. elegans*. For example, spinning disk confocal microscope model, objective maker, Z-stack interval, exposure time or pixel dwell time, time-lapse interval and duration. And did the authors show z max-projection image in Figure 2? Any image processing also needs to be described.

*Reviewer #3 (Recommendations for the authors):*

– Doublecheck grammar and spelling (e.g., "themself" in line 25).

– Consider rewriting some parts to avoid ambiguity (e.g., 10 mRNAs or mRNAs from 10 genes in the line 43?).

– A bit more expanded introduction would help readers to appreciate the importance of the authors' work.

– In the Methods, the live imaging section needs more details such as the *C. elegans* immobilization method.

– The authors need to be clear on what version of MS2 stem loops they used as a different version can make different results. For example, the version 5 can induce clustering of MCP signals even after the target mRNAs are degraded due to its nature of being stable and aggregated in the cell.

*Reviewer #4 (Recommendations for the authors):*

1. While the manuscript provides ample details on constructs, the information on *C. elegans* imaging is minimal. The authors should provide z-stack information (how many planes and step size), temporal resolution, camera used and settings, laser settings, etc.

2. Analysis of foci sizes. Naively, one would expect quantile sizes of the foci with one, two, three, … mRNAs in a dot, although the apparent/measured size increase may not be linear. Figure 2F, which shows foci sizes over time, looks puzzling in this context.

a. How should one interpret the initial ramp up (first 2 minutes) before hitting an apparent plateau? Could it be an artifact of measurement?

b. Given the assumption of quantal sizes, the plotted mean +/- sem can be misleading. A scatter plot of individual size would make more sense. Alternatively, the authors could provide the size distribution at some representative time points (eg, the two plateaus 3-6 min and 10-15).

c. Related, to what extent could one infer the unit size/single mRNA signal? Some of the zoomed in images leave the impression that there are a large number of foci with comparable and smallest size, which could be single mRNAs?

d. What is the y-axis? I assume it's diameter in um? Since the sizes are quantal in volume, would it make more sense to make some of the plots in d^3^?

3. Some missed opportunities on the extent of usefulness of the current reagents. For example:

a. Is it possible to see signals in nuclei to capture transcription and early posttranscriptional regulation?

b. What's the practical depth penetrance of imaging? Could one get reasonable signal at the deep side (basal side) of the epidermal cells? Or the far side of the worm (different larval stages provide different distances for imaging).

4. Some phenotypic assays on the physiological relevance/potential interference of abelling could greatly strengthen the usefulness of the systems. Some suggestions for the authors to consider:

a. Labeling of cdc-42: Does it affect the actomyosin dynamics in wound repair? Does it affect any developmental process (eg, post-embryonic cell divisions)?

b. Functional rescue by the tagged allele (heterozygous over null/loss of function allele) on any of the three genes tagged?

5. Some analysis and discussions on the range of optimization would be very useful for the field.

a. What is the signal to background ratio? With gross simplification/linear extrapolation, could one consider reducing the abelling (number of GFP recruited) by 10x, 20x or 50x reduction of amplification?

b. How practical is the imaging setting in terms of phototoxicity and longer term imaging? If the authors could provide a powermeter readout of laser after the lens, it would be most objective and transferable to other labs. If not, a more detailed documentation of the laser used and settings would help.

6. I expect the community would respond enthusiastically to the publication and there would be a lot of interest on the reagents. Given the open science spirit of *eLife*, the authors should affirm the deposition of constructs and worm strains to public platforms, eg, Addgene and CGC.

---

## [Author Response]

Essential revisions:The authors have not compared their MASS system to a standard in the field, the MS2 system. There are questions on the relevance of this approach for single molecule imaging, and the effect of the tagging on the physiology of the mRNA. There needs to be a more rigorous approach showing that the mRNA tagged in this way behaves as if it was not tagged. Suggestions also include comparison to FISH images in fixed cells, and measurement of velocities with and without the extensive tagging. It may be that the massive amount of protein on the mRNA may render it inert.The referencing does not include similar work in the field (eg Guo et al) that needs to be discussed and compared more thoroughly.Provide quantitative evaluation of signal and increase in signal to noise. Refer to the reviewers’ many comments and suggestions.

We thank the editors for spending time with our manuscript and encouraging comments. We have done FISH experiments, performed more quantitative evaluations of signal-tonoise, and signal intensity, examined mRNA stability, and measured the velocities with a conventional MS2 system and MASS.

Reviewer #2 (Recommendations for the authors):1. Line 44: There are more previous works reporting endogenous mRNA imaging in live animals.Forrest and Gavis, “Live imaging of endogenous RNA reveals a diffusion and entrapment mechanism for nanos mRNA localization in *Drosophila*”, Current Biology, 13, 1159 (2003)

We thank the reviewer for providing more references. We have added the missing

references in the new version of the manuscript on page 2.

2. Line 410: More details need to be described for live imaging in *C. elegans*. For example, spinning disk confocal microscope model, objective maker, Z-stack interval, exposure time or pixel dwell time, time-lapse interval and duration. And did the authors show z max-projection image in Figure 2? Any image processing also needs to be described.

We updated the methods section and the missing information on page 25.

Reviewer #3 (Recommendations for the authors):– Doublecheck grammar and spelling (e.g., “”I" in line 25).– Consider rewriting some parts to avoid ambiguity (e.g., 10 mRNAs or mRNAs from 10 genes in the line 43?).

We apologize for the ambiguity. We have checked grammar and spelling and c“anged

"less than t”n mR“As" to "mRNAs from less than t”n genes".

– A bit more expanded introduction would help readers to appreciate the importance of the’authors' work.

We noticed there are several beautiful review articles about RNA imaging published recently. We, therefore, cited those papers and wrote a short introduction.

– In the Methods, the live imaging section needs more details such as the *C. elegans* immobilization method.

We have now updated the methods section, and the missing information was added to page 26.

– The authors need to be clear on what version of MS2 stem loops they used as a different version can make different results. For example, the version 5 can induce clustering of MCP signals even after the target mRNAs are degraded due to its nature of being stable and aggregated in the cell.

We used the V1 version of MS2 stem-loop in our study. We added this information to the manuscript. We also tested the MS2 V7, an optimized version of the MS2 stem-loop, and found that MASS also works with MS2 V7. We added this new data in Figure 1-figure supplement 2.

Reviewer #4 (Recommendations for the authors):1. While the manuscript provides ample details on constructs, the information on *C. elegans* imaging is minimal. The authors should provide z-stack information (how many planes and step size), temporal resolution, camera used and settings, laser settings, etc.

We updated the methods section, and the missing information was added to page 26.

2. Analysis of foci sizes. Naively, one would expect quantile sizes of the foci with one, two, three, … mRNAs in a dot, although the apparent/measured size increase may not be linear. Figure 2F, which shows foci sizes over time, looks puzzling in this context.a. How should one interpret the initial ramp up (first 2 minutes) before hitting an apparent plateau? Could it be an artifact of measurement?

We imaged exogenous BFP-8xMS2 mRNA in the epidermis of *C. elegans*. We found that the size of GFP foci of BFP-8xMS2 mRNA kept constant over time. We also found that the size of endogenous C42D4.3-8xMS2 mRNAs is bigger than the GFP foci of BFP-8xMS2 mRNAs. Those data indicate that the growing size of the GFP foci of endogenous C42D4.3 mRNA is not an artifact of measurement.

Our data suggested that there is a transcription boost in the first two minutes after wounding, and those mRNAs undergo quick clustering to form RNA granules.

b. Given the assumption of quantal sizes, the plotted mean +/- sem can be misleading. A scatter plot of individual size would make more sense. Alternatively, the authors could provide the size distribution at some representative time points (eg, the two plateaus 3-6 min and 10-15).

We made the scatter plot of the individual size of each GFP foci at five representative time points (1 min, 2 min, 3 min, 4 min, and 5 min after wounding). We added that information to Figure 2—figure supplement 5.

c. Related, to what extent could one infer the unit size/single mRNA signal? Some of the zoomed in images leave the impression that there are a large number of foci with comparable and smallest size, which could be single mRNAs?

There are indeed small and large foci of mRNAs in cells. We imaged exogenous BFP8xMS2 mRNAs in the epidermis of *C. elegans* and found that the size of the GFP foci of endogenous C42D4.3-8xMS2 mRNAs is larger than that of BFP-8xMS2 mRNAs. Those data suggest that the large GFP foci in C42D4.3-8xMS2 mRNA are mRNA granules. GFP foci with the smallest size should be single mRNAs. We added those new data in Figure 2—figure supplement 5 and the text on page 7.

We also performed MASS combined with single-molecule RNA FISH. We found that MASS detected a similar number of GFP foci compared to the spots detected by smFISH. In addition, the majority (72%) of GFP foci colocalized with the smFISH spots of b-ACTIN8xMS2 mRNAs. Thus These data suggested that MASS could label single mRNA

molecules, and GFP foci with minimal size are likely single mRNAs. We have added the new data in Figure 1, Figure 1—figure supplement 1, and the text on page 3.

d. What is the y-axis? I a’sume it's diameter in um? Since the sizes are quantal in volume, would it make more sense to make some of the plots in d^3^?

The y-axis is the area (µm^2^) of the GFP foci. We added this missing information in Figure 2 and Figure 2—figure supplement 5.

3. Some missed opportunities on the extent of usefulness of the current reagents. For example:a. Is it possible to see signals in nuclei to capture transcription and early posttranscriptional regulation?

We expected a boost of transcription after wounding. However, we failed to detect the appearance of bigger GFP foci in the nucleus. The epidermis of *C. elegans* is a syncytium with 139 nuclei located in different focal planes. With our microscopy, we were able to image only one focal plane, in which there are usually only four to ten nuclei. Therefore, it is likely that the nuclei with active transcription were out of focus. We discussed this point in the manuscript (page 6).

b. What's the practical depth penetrance of imaging? Could one get reasonable signal at the deep side (basal side) of the epidermal cells? Or the far side of the worm (different larval stages provide different distances for imaging).

The practical depth penetrance of our spinning microscopy is 50 µm. The depth of the epidermis of *C. elegans* is 60 µm. Therefore we are only able to image the basal side of the epidermal cell.

4. Some phenotypic assays on the physiological relevance/potential interference of labeling could greatly strengthen the usefulness of the systems. Some suggestions for the authors to consider:a. Labeling of cdc-42: Does it affect the actomyosin dynamics in wound repair? Does it affect any developmental process (eg, post-embryonic cell divisions)?

We tagged 8xMS2 into the 3′UTR of cdc-42 mRNA. The coding sequence of cdc-42 mRNA was not changed. We previously found that epidermal wounding induces rapid clustering of CDC-42 proteins. When cdc-42-8xMS2 mRNAs were overexpressed and labeled by MASS in the epidermis of *C. elegans*, we found similar clustering of CDC-42 proteins, suggesting MASS did not affect the function of CDC-42. Therefore, the actomycosin dynamics in wound repair were likely not affected. We did not observe any developmental defects in worms expressed with cdc-42-8xMS2 mRNAs.

b. Functional rescue by the tagged allele (heterozygous over null/loss of function allele) on any of the three genes tagged?

We tagged 8xMS2 into the 3′UTR of endogenous C42D4.3 and mai-1 mRNAs. The coding sequences of C42D4.3 and mai-1 mRNAs were not changed. We showed that MASS did not affect the expression level and mRNA stability of C42D4.3 gene. We expect the function of C42D4.3 and mai-1 will not be affected, and we did not observe any developmental defects in the tagged worms. Therefore we did not perform functional rescue experiments by the tagged allele.

5. Some analysis and discussions on the range of optimization would be very useful for the field.a. What is the signal to background ratio? With gross simplification/linear extrapolation, could one consider reducing the labeling (number of GFP recruited) by 10x, 20x or 50x reduction of amplification?

For the signal-to-noise ratio, we did more experiments and analyses. We imaged overexpressed b-ACTIN mRNAs using the conventional 24xMS2 system or MASS with different repeats of Suntag arrays (MCP-24xSuntag, MCP-12xSuntag, MCP-6xSuntag). For the conventional 24xMS2 system, we followed the previous protocol that added a nuclear localization signal (NLS) to MCP, and b-ACTIN mRNAs were nicely detected with a signal-to-noise ratio of 1.21.

We found MASS showed a comparable or better signal-to-noise ratio than the conventional 24xMS2 system. (MASS with MCP-24xSuntag: 1.79, MASS with MCP-12xSuntag: 1.48, MASS with MCP-6xSuntag: 1.42). These data indicate that using Suntag as a signal amplifier did not increase background noise.

b. How practical is the imaging setting in terms of phototoxicity and longer term imaging? If the authors could provide a powermeter readout of laser after the lens, it would be most objective and transferable to other labs. If not, a more detailed documentation of the laser used and settings would help.

We do not have a powermeter readout of laser after the lens. We updated the methods section, and the information on the laser used and settings were added to page 26.

We did not observe significant phototoxicity in the *C. elegans* after 15 min continuous imaging with a time interval of 2 seconds.

We did not test whether MASS could be used for longer-term imaging in this study. However, it is recently published that SunRISER, which used a similar strategy with MASS, allows long-term (24 h with a 10 min frame rate) live cell mRNA imaging (Guo, Y. and Lee, R.E.C., Cell Reports Methods, 2022). Taking together, Guo and we demonstrated that it is an efficient strategy to combine the MS2 system and the Suntag system as a signal amplifier for long-term and endogenous mRNA imaging in live cells.

6. I expect the community would respond enthusiastically to the publication and there would be a lot of interest on the reagents. Given the open science spirit of eLife, the authors should affirm the deposition of constructs and worm strains to public platforms, eg, Addgene and CGC.

Thanks for the suggestion; yes, we will deposit all the constructs and worm strains to

Addgene and CGC after this paper's publication.